# Spatial–Temporal Heterogeneity and Driving Factors of Rural Residents' Food Consumption Carbon Emissions in China—Based on an ESDA-GWR Model

**Shuai Qin** **, Hong Chen * and Haokun Wang**

College of Economics and Management, Northeast Forestry University, Harbin 150040, China;
15225145321@nefu.edu.cn (S.Q.); whk@nefu.edu.cn (H.W.)
* Correspondence: chenhong@nefu.edu.cn; Tel.: +86-139-3644-1436

**Abstract:** The increase in income among Chinese residents has been accompanied by dramatic changes in dietary structure, promoting a growth in carbon emissions. Therefore, in the context of building a beautiful countryside, it is of great significance to study the carbon emissions of rural residents' food consumption to realize the goal of low-carbon food consumption. In this paper, the calculation of food consumption carbon emissions of Chinese rural residents is based on the carbon conversion coefficient method, and the spatial heterogeneity of influencing factors is analyzed with the aid of the ESDA-GWR model. The results indicate that the per capita food consumption carbon emissions of rural residents have increased by 1.68% annually, reaching 336.73 kg $CO_2$-eq in 2020, which is 1.32 times that of 2002. Carbon emissions generated from rural residents' food consumption have significant spatial agglomeration characteristics, showing the spatial distribution characteristics of a north–south confrontation, with a central area collapse. The influencing factors of food consumption carbon emissions have significant spatial heterogeneity, among which, as the main force to restrain the growth of food consumption carbon emissions, the price factor has a regression coefficient between −0.1 and −0.3, and its influence has weakened from northwest to southeast in 2020. The education–social factor is the main driving force for the growth of food consumption carbon emissions, with a regression coefficient between 0.58 and 0.99, and its influence has increased from east to west. In the future, formulating food consumption optimization policies should be based on the actual situation of food consumption carbon emissions in various regions to promote the realization of low-carbon food consumption.

**Keywords:** food consumption carbon emissions; spatial–temporal heterogeneity; geographically weighted regression; food low-carbon consumption

## 1. Introduction

In 2006, China became the largest emitter of carbon dioxide in the world [1,2]. In order to achieve the United Nations Sustainable Development Goals (SDGs) and reduce the adverse effects of carbon emissions, the Chinese government made promises to reduce carbon emissions at the 75th United Nations General Assembly, commit to achieving the peak of carbon emissions in 2030, and strive to achieve the strategic development goal of carbon neutrality in 2060. China is currently making many efforts to achieve this goal [3]. As economic development enters a new normal, China's production and consumption structure has undergone major changes [4–6]. Consumption carbon emissions have been increasing and mainly come from the household sector [7,8]. Among them, food consumption is not only a significant part of household consumption, but also an important source of greenhouse gases [9]. Previous studies have shown that 19–29% of human-made greenhouse gas emissions stem from food consumption [10,11]. This problem is more prominent in China because China has the burden of feeding its 1.4 billion people [12]. As the world's largest food producer and consumer, China's food carbon

emissions are higher than any country in the world [13]. Unreasonable dietary consumption creates huge pressure on the ecological environment [14]. Reducing carbon emissions due to the structural evolution of food consumption has become an important issue for Chinese scholars in the context of building a beautiful countryside and high-quality agricultural development [15–17].

The research on food consumption carbon emissions is divided into two components; the first is the calculation of food consumption carbon emissions and the second is the analysis of influencing factors. Previous studies have shown that the calculation of food consumption carbon emissions is mainly divided into a production perspective and a consumption perspective [13,18], among which major scholars conduct research based on the production perspective, and relatively few researchers examine the consumption perspective [19]. It is worth noting that the calculation of food consumption carbon emissions from the consumption perspective can be divided into direct carbon emissions and indirect carbon emissions. Food consumption direct carbon emissions refers to the carbon emissions from the consumption of food itself [19,20]. Using the carbon conversion coefficient method for calculation, Cao et al. (2020) found that the average annual increase in per capita food consumption carbon emissions was 1.68% in China [21]. Studies of indirect food consumption carbon emissions focus on the carbon emissions of the entire food chain (food processing, production, transportation, and storage stages) from the perspective of food life cycles [12,22,23]. This calculation is mainly based on the life cycle method and the input–output method. For example, based on the input–output method, Feng et al. (2020) found that, compared to 1992, China's per capita food consumption carbon emissions were reduced by 21% in 2007 [9]. Yang et al. (2019) found that Chinese residents' food consumption carbon emissions were 683.38 g $CO_2$-eq per day per capita based on the life cycle method, noting that optimizing dietary structure can reduce carbon emissions by 40% [24]. However, a comprehensive study on food consumption carbon emissions in China has not yet been completed [12], especially from a rural residents' perspective. Current research mainly focuses on the national level or only involves urban residents [4,7,9,22,23], ignoring rural residents' food consumption carbon emissions. However, even if China's population reaches a peak of 1.45 billion by 2030, 30% of the population will still live in rural areas, which is about 4.35 billion people; currently, this population's meat consumption is almost the same as that of urban residents, about 77 kg per capita per year [25]. Additionally, the carbon emissions for per unit meat consumption are much higher than those of plant-based foods [26]. Therefore, it is necessary to conduct an in-depth analysis of rural residents' food consumption carbon emissions, especially in the context of the country's vigorous promotion of building a beautiful countryside and high-quality agricultural development.

Reducing the impact of carbon emissions on the ecological environment is an important part of achieving the 13th Sustainable Development Goal. D'Adamo et al. (2021) found that the application of bioenergy technology to food production is vital to human society and the key to achieving a low-carbon society [27]. Vacchi et al. (2021) expounded the importance of technological progress for sustainable development, defined the concept of technological sustainability, and developed a technological sustainability assessment framework based on the organization and production perspective of the manufacturing industry [28]. Mies et al. (2021) hold that most of the current research on sustainability focuses on economic and ecological sustainability, ignoring the study of social sustainability dimensions, which is not conducive to the circular development of the entire economy, and used causal loop modelling to explore the cross relationship between social sustainability and economic and ecological sustainability [29]. The above-mentioned scholars have explained in detail the different dimensions of sustainable development. For us, finding the reasons for changes in food consumption carbon emissions is the key to reducing food consumption carbon emissions in order to achieve the Sustainable Development Goals [12]. At present, structural decomposition analysis (SDA) and index decomposition analysis (IDA) are often used to analyze the influencing factors of food consumption carbon emissions [8,30]. They analyze the impact of technological advancement, dietary



structure evolution, energy intake, population scale, residents' income, and urbanization on residents' food consumption carbon emissions, with technological progress being the main factor restraining the increase in carbon emissions from food consumption; however, the evolution of dietary structure has become the main reason for the increase in carbon emissions from food consumption in recent years [8,9,20,30,31]. Previous studies have laid a solid foundation for the development of this article. However, the above research can only reveal the regional average value of the various factors on food consumption carbon emissions. There has heretofore been an inability to reveal the heterogeneity of various factors in different regions, as China is vast and the dietary structure of residents in different regions is quite different [21]. Therefore, it is important to conduct a detailed analysis of the regional heterogeneity of each factor in order to provide a theoretical reference for the formulation of low-carbon food consumption policies in different regions.

Under such circumstances, the main aim of this work is to analyze the spatiotemporal evolution trends in carbon emissions generated from rural residents' food consumption, and analyze the spatial heterogeneity of its influencing factors, so as to provide a reference for the formulation of differentiated low-carbon food consumption policies. The specific objectives consist of two parts. The first is to calculate the carbon emissions from rural residents' food consumption in 31 provinces from 2002 to 2020, based on the carbon conversion coefficient method, and further analyze the emissions' spatial distribution characteristics. The second approach is to select the influencing factors of food consumption carbon emissions on the basis of previous studies and study the spatial heterogeneity of each factor based on the geographically weighted regression model (GWR). Although several articles have studied the carbon emissions of Chinese residents' food consumption, the marginal contribution of this article lies in the following: (1) filling the research gap regarding the carbon emissions generated by rural residents' food consumption (previous studies have often used the whole country as the research object or only analyzed urban residents [4,7,9,23], ignoring the huge food consumer group of rural residents); (2) rather than obtaining an average value of influencing factors based on a decomposition model, studying the spatial heterogeneity of each factor based on a geographically weighted regression model (GWR); and (3) adopting more recent data published by the National Bureau of Statistics of China (until 2020). These contributions can provide a valuable theoretical basis for policymakers to formulate differentiated food low-carbon consumption policies, thereby achieving sustainable food consumption goals.

The remaining content of this article is arranged as follows. Section 2 is model construction and data sources. Section 3 lists the research results from two aspects: One is the study of the temporal and spatial distributions of carbon emissions from food consumption among rural residents, and the other an analysis of the spatial heterogeneity of the influencing factors of carbon emissions from food consumption. Discussions and conclusions are found in Sections 4 and 5, respectively.

## 2. Model Specification and Data Source

### 2.1. Calculation Method of Per Capita Food Consumption Carbon Emissions

In order to accurately measure rural residents' per capita food consumption carbon emissions in China, based on previous studies [32], the carbon conversion coefficient method was used. The specific formula is as follows:

$$TC_j = \sum_{i=1}^{j} C_{ij} = \sum_{i=1}^{j} FC_{ij} \times r_i \tag{1}$$

where $TC_j$ is the per capita food consumption carbon emissions of rural residents in the $j$ area; $C_{ij}$ refers to the carbon emissions of the $i$ food consumption of residents in the $j$ area; $FC_{ij}$ refers to the $i$ food consumption of the residents in the $j$ area; $r_i$ is the carbon emission coefficient of the $i$ food; $i$ is the type of food consumption, with a value of 12; and $j$ is the number of provinces (cities, districts), with a value of 31. The carbon emission factor of

various types of food comes from Tilman (2014) [26]. Since the unit of the carbon emission coefficient in the literature is g/kcal (1 kcal = 4186 J), the energy conversion rate of various foods needs to be used for correction [21]. The carbon emission coefficients of various foods are shown in Table 1.

**Table 1.** Carbon emission coefficients of various types of foods (kg $CO_2$-eq/kg).

| Food Type | Grain | Vegetable | Vegetable Oil | Sugar | Fruit | Pork |
|---|---|---|---|---|---|---|
| Coefficient | 0.27 | 0.40 | 1.48 | 0.08 | 0.07 | 7.64 |
| **Food Type** | **Beef** | **Lamb** | **Poultry** | **Egg** | **Milk** | **Aquatic Products** |
| Coefficient | 12.04 | 18.86 | 1.71 | 0.78 | 0.36 | 1.94 |

### 2.2. Exploratory Spatial Data Analysis

The core content of the exploratory spatial data analysis (ESDA) method is spatial autocorrelation detection, which mainly includes global spatial autocorrelation and local spatial autocorrelation [33].

Global spatial autocorrelation is the description of the spatial characteristics of the attribute value of a variable under the same spatial unit to determine whether there is an agglomeration effect in space. The formula is:

$$I = \frac{n \sum\limits_{i=1}^{n} \sum\limits_{i=1}^{n} w_{ij} \left( x_i - \overline{x} \right) \left( x_j - \overline{x} \right)}{\sum\limits_{i=1}^{n} \sum\limits_{j=1}^{n} w_{ij} \sum\limits_{i=1}^{n} \left( x_i - \overline{x} \right)^2} \tag{2}$$

where $I$ is the global Moran's index; the value range is [–1, 1]; $I$ greater than zero means a spatial positive correlation; $I$ less than zero means a spatial negative correlation; $I$ equal to zero means a random distribution with no spatial correlation; $n$ equal to 31 refers to the number of spatial units in the study area; $w_{ij}$ represents the spatial weight matrix; if $i$ and $j$ are adjacent, then $w_{ij}$ is equal to 1, otherwise, it is 0; $x_i$ and $x_j$ are the observed values of regions $i$ and $j$; and $x$ is the mean value.

Local spatial autocorrelation makes up for the lack of global spatial autocorrelation, which is mainly used to reveal the spatial heterogeneity between a certain area and the surrounding area. The commonly used methods are the Local Moran's $I$ index and the Getis–Ord $G_i^*$ index. In order to better reflect the degree of clustering of high and low values in a local area, this study uses the Getis–Ord $G_i^*$ index to measure the cold and hot agglomeration areas of rural residents' per capita food consumption carbon emissions. The formula is:

$$G_i^* = \sum\limits_{j=1}^{n} W_{ij} \times x_j / \sum\limits_{j=1}^{n} x_j \tag{3}$$

$$Z\left(G_i^*\right) = \frac{G_i - E(G_i)}{\sqrt{Var(G_i)}} \tag{4}$$

where $E\left(G_i\right)$ and $Var\left(G_i\right)$ are the expected value and variance of $G_i$. A significant positive value of $Z\left(G_i^*\right)$ indicates that the value around the $i$ area is relatively high, i.e., high-value space clusters are hot spots, and vice versa, whereas low-value space clusters are cold spots.

### 2.3. Geographically Weighted Regression

Different from the traditional regression model (OLS), the geographically weighted regression (GWR) model incorporates spatial location information in the regression equa-

tion, which can be used to reflect the non-stationarity of parameters at different spatial locations [34]. Its model structure is

$$y_i = \beta_0(u_i, v_i) + \sum_k \beta_k(u_i, v_i)x_{ik} + \varepsilon_i \tag{5}$$

where $y_i$ refers to the value of the explained variable at the geographic location $(u_i, v_i)$, $\beta_0 (u_i, v_i)$ represents the constant value at the geographic location $(u_i, v_i)$, $(u_i, v_i)$ refers to the geographic center coordinates of the sample space unit, $\beta_k (u_i, v_i)$ is the value of the function $\beta_k (u, v_i)$ in the space position of the $i$ sample, and $\varepsilon_i$ represents the spatial random residual.

### 2.4. Data Sources

In this paper, the types of rural residents' food consumption include 12 categories, namely, grains, vegetables, vegetable oil, sugar, fruit, pork, beef, lamb, poultry, eggs, milk, aquatic products, etc. The data mainly come from the China Statistical Yearbook, China Rural Statistical Yearbook, and China Rural Household Survey Yearbook. The per capita disposable income, per capita GDP, food consumption expenditure, consumer price index, food retail price index, and Engel coefficient are all from the China Statistical Yearbook. Among them, per capita disposable income, per capita GDP, and food consumption expenditure are deflated based on the CPI index in 2002. The dietary structure uses the ratio of animal-based food consumption to plant-based food consumption as a proxy variable. The per capita education level is calculated based on a method by Chen (2004) [35].

## 3. Results and Discussion

### 3.1. Temporal Evolution Characteristics of Rural Residents' Per Capita Food Consumption Carbon Emissions

The per capita food consumption and carbon emissions of rural residents in China are calculated based on the carbon conversion coefficient method and Formula (1) (Figure 1).

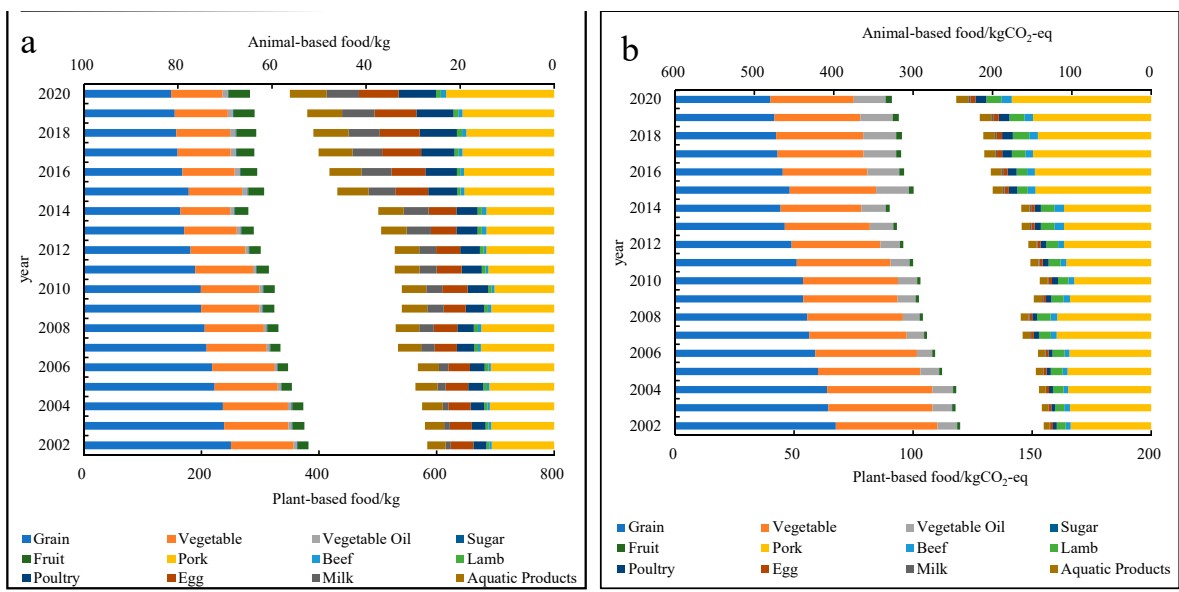

**Figure 1.** Per capita food consumption (**a**) and carbon emissions among rural residents (**b**) from 2002 to 2020.

It can be seen from Figure 1a that during the study period, per capita food consumption decreased by 17.16%, from 409.00 kg in 2002 to 338.80 kg in 2020. Per capita plant-food consumption still accounted for 83.14% in 2020, and although it decreased by 1.61% annually, it was still the main type of food consumption. Per capita consumption of animal-based food maintained a higher growth rate at about 4.32%, however, the amount of consumption was still small. Since the carbon emissions in producing animal food are higher than those

of plant-based food [21,26], the transformation of the food consumption structure of rural residents has promoted continuous changes in food consumption carbon emissions.

According to Figure 1b, per capita food consumption carbon emissions of rural residents increased by 1.68% annually, reaching 336.73 kg $CO_2$-eq by 2020, which is 1.32 times that of 2002. The per capita carbon emissions of animal-based food were always higher than those of plant-based food. Specifically, per capita carbon emissions from plant-based foods decreased to 91.06 kg $CO_2$-eq in 2020 at an average annual growth rate of 1.5%. Among them, per capita carbon emissions of grains and vegetables were reduced by 2.80% and 1.05%, respectively, at an average annual growth rate. The average annual growth rate of per capita carbon emissions of vegetable oil, sugar, and melons and fruits increased by 3.35%, 0.47%, and 4.10%, respectively. The average annual per capita carbon emissions of animal-based foods increased by 3.60%, from 135.44 kg $CO_2$-eq in 2002 to 245.67 kg $CO_2$-eq in 2020. Among them, the per capita carbon emissions of dairy foods showed the fastest growth rate, with an average annual growth rate of 11.68%, followed by poultry at 6.44%, beef at 5.95%, aquatic products at 4.08%, mutton at 3.67%, pork at 3.54%, and eggs at 3.40%. Pork, as the main source of animal-based food for residents, had a far lower emission growth rate than milk. The main reason for this is that rural residents paid more attention to health problems and increased their intake of low-fat and high-protein foods with improvements in income and education level.

To summarize, plant-based food is still the main source of food among rural residents, but rapid changes in dietary structure have led to a continuous increase in per capita carbon emissions. Animal-based food is the main food type that promotes an increase in per capita carbon emissions.

### 3.2. Spatial Distribution Characteristics of Rural Residents' Per Capita Food Consumption Carbon Emissions

In the context of a continuous increase in rural residents' per capita food consumption carbon emissions, we used ArcGIS 10.8 software to divide the rural residents' per capita food consumption carbon emissions in each province into four levels from low to high; this was conducive to spatial analysis and expression. In order to save space, only beginning and ending years are reported (Figure 2).

Per capita food consumption carbon emissions of rural residents showed significant regional differences within the research span. As a whole, the research presented the distribution characteristics of a north–south confrontation, with a central area collapse. Higher carbon emission provinces for per capita food consumption dropped from 10 in 2002 to 7 in 2020, namely, Inner Mongolia, Qinghai, Sichuan, Chongqing, Guangdong, Fujian, and Shanghai, among which Guangdong, Fujian, and Shanghai are located in the eastern coastal zone of China and are relatively developed. The improvements in their respective economic levels are accompanied by a higher intake of animal-based foods [36]. Per capita food consumption carbon emissions will continue to stay at a relatively high level as the economy develops because the carbon emission coefficient of animal-based food is relatively high. In addition, Inner Mongolia, Qinghai, Sichuan, and Chongqing are not only areas in which ethnic minorities live in concentrated communities in China, but they are also the areas in which animal-based food is consumed more regularly. In the study interval, the average per capita consumption of animal-based food was higher than the national level. Therefore, the per capita carbon emissions of food consumption in the above seven provinces are always in high-value areas. Low-carbon-emission provinces in terms of per capita food consumption rose from five in 2002 to six in 2020, namely, Hebei, Henan, Shandong, Shanxi, Shaanxi, and Gansu. These regions are distributed in the central region of China. The level of economic development is not only inferior to Guangdong, Fujian, and Shanghai but also lacks economic support for animal-based food consumption. Compared with Inner Mongolia, Qinghai, Sichuan, and Chongqing, these regions are more affected by traditional Chinese vegetarian culture; they consume more plant-based foods and fewer animal-based foods, leading to the per capita food consumption carbon emissions being at a relatively low level in these areas. As a result,

per capita food consumption carbon emissions of rural residents in China show the spatial distribution characteristics of a north–south confrontation, with a central area collapse.

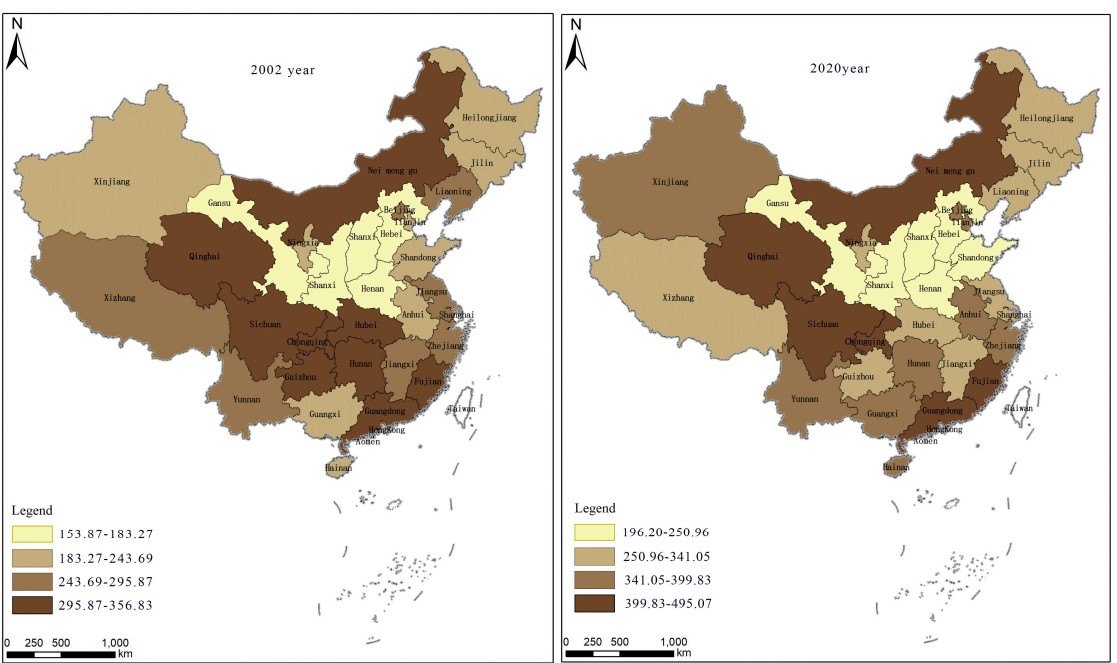

**Figure 2.** The spatial distribution of various provinces' rural residents' per capita food consumption carbon emissions from 2002 to 2020/(kg CO$_2$-eq).

### 3.3. Spatial Autocorrelation Analysis of Rural Residents' Per Capita Food Consumption Carbon Emissions in China

In order to analyze the spatial correlation of rural residents' per capita food consumption carbon emissions in 31 provinces from 2002 to 2020, we used Formula (2) and the spatial autocorrelation statistical tool from the ArcGIS 10.8 software to calculate the Global Moran's *I* Index (Table 2). The Global Moran's *I* index shows that per capita food consumption carbon emisions from 2002 to 2020 were all positive, and they all passed the test at a significance level of 10%, except for in 2003. These values indicate that the rural residents' per capita food consumption carbon emissions in 31 provinces have strong spatial agglomeration characteristics, i.e., one province's per capita food consumption carbon emissions are positively affected by neighboring regions.

In order to better reflect the degree of high and low clustering values in a local area, the $G_i$* index was calculated by ArcGIS 10.8 software according to Formulas (3) and (4). Additionally, it was divided into four grades from low to high based on the natural break point analysis method (Jenks), among which a high GiZScore represents hot spots with high carbon emissions, and vice versa for cold spots. Only 2002 and 2020 were selected for visual expression for the purpose of saving space (Figure 3). We found that the hot spots were mainly distributed in southern China from 2002 to 2020, indicating that the coupling degree between residents' food consumption and ecological environment was low and urgently needed to be improved in this region. The cold spots were mostly concentrated in the northern region, where the coupling degree between residents' food consumption and the ecological environment was relatively high, but the number was relatively small. From the perspective of development trends, the number of hot spots in 2020 decreased by three compared to 2002. There was also a trend towards the eastern coast, reflecting that low-carbon food consumption in eastern coastal areas is in a bottleneck period. This produces two issues: The endogenous power of low-carbon food consumption is insufficient, and it is difficult to receive exogenous radiation from the advanced northern regions. External policies are needed to prevent the hotspot from spreading to the area. The number of

cold spots increased from five in 2002 to six in 2020, and were mainly concentrated in the northern region. The increase in this number is conducive to the positive spatial spillover effect of low-carbon regions. During the study period, five provinces, including Inner Mongolia, Hebei, Shanxi, Henan, and Shandong, continued to stay in low-value areas and showed obvious agglomeration effects, indicating that the coupling degree of rural residents' food consumption and ecological environment in these five provinces was relatively high and could be used as a national level benchmark in cities to promote the realization of low-carbon food consumption in other provinces.

**Table 2.** Spatial autocorrelation results of rural residents' per capita food consumption carbon emissions in China from 2002 to 2020.

| Year | Moran's I Value | Z-Value | *p*-Value |
|------|------------------|---------|-----------|
| 2002 | 0.266 | 2.717 | 0.007 |
| 2003 | 0.261 | 2.681 | 0.007 |
| 2004 | 0.241 | 2.497 | 0.013 |
| 2005 | 0.114 | 1.398 | 0.162 |
| 2006 | 0.159 | 1.755 | 0.079 |
| 2007 | 0.259 | 3.014 | 0.003 |
| 2008 | 0.290 | 2.934 | 0.003 |
| 2009 | 0.228 | 2.374 | 0.018 |
| 2010 | 0.175 | 1.893 | 0.058 |
| 2011 | 0.155 | 1.709 | 0.088 |
| 2012 | 0.221 | 2.374 | 0.018 |
| 2013 | 0.245 | 2.613 | 0.009 |
| 2014 | 0.227 | 2.422 | 0.015 |
| 2015 | 0.278 | 2.860 | 0.004 |
| 2016 | 0.307 | 3.266 | 0.001 |
| 2017 | 0.273 | 2.942 | 0.003 |
| 2018 | 0.219 | 2.429 | 0.015 |
| 2019 | 0.194 | 2.073 | 0.038 |
| 2020 | 0.170 | 1.855 | 0.064 |

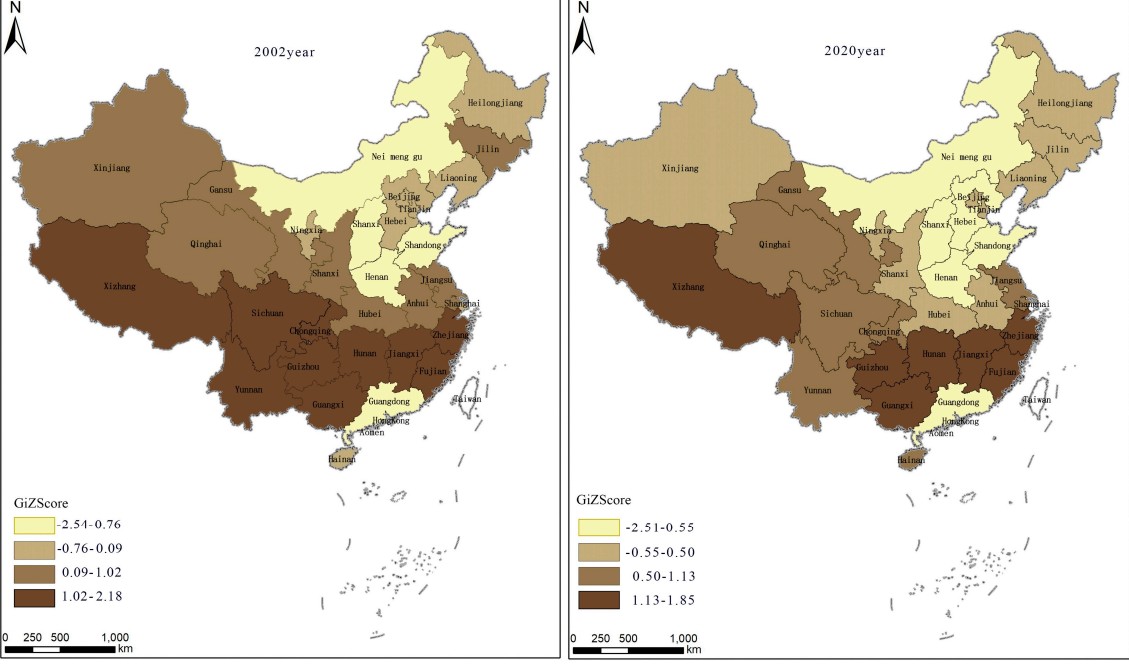

**Figure 3.** Distribution of cold and hot spots of rural residents' per capita food consumption carbon emissions in China from 2002 to 2020.

*3.4. Effective Factor Analysis of Rural Residents' Per Capita Food Consumption Carbon Emissions in China*

3.4.1. Selection of Influencing Factors

Per capita food consumption carbon emissions are affected by multiple factors. Scholars generally believe that per capita disposable income, food consumption expenditure, the Engel coefficient, and other factors all have an effect on emissions [20,23,32]. Based on the research purpose and the availability of the data, nine indicators, including per capita disposable income and per capita GDP, were selected as the influencing factors of per capita food consumption carbon emissions. The principal component analysis method was used to process the original data for the purpose of eliminating the multicollinearity between variables. We found that the KMO values in 2002 and 2020 were 0.62 and 0.65, respectively, both greater than 0.60. The Bartlett sphere test values were 235.37 and 216.30, respectively, and the sig values were both 0.00. The cumulative variance contribution rates were 85.60% and 83.28%, respectively, indicating that the rotated factor retained more original information and met the analysis requirements. Four principal components were extracted based on the principle that the characteristic root was greater than 1, and these were named the economic–preference factor, the education–social factor, the material factor, and the price factor (Table 3). The economic–preference factor represents the economic foundation and dietary preferences of rural residents. The education–social factor refers to the diversification of residents' cultural level and consumption choices. The material factor reflects food availability and the suitability of the local commodity market. The price factor represent the sensitivity of residents to food prices.

**Table 3.** Driving factors and variable explanation.

| Driving Factors | Variable System | Variable Interpretation |
|---|---|---|
| Economic–preference factor | Per capita disposable income | Reflects the income level of rural residents |
| | Per capita GDP | Reflects the level of economic development in rural areas |
| | Food consumption expenditure | Characterizes the cash input of rural residents in food consumption |
| | Dietary structure | Represents the food consumption preferences of rural residents |
| Education–social factor | Per capita average years of education | Reflects the cultural quality of rural residents |
| | Engel coefficient | Reflects the diversity of consumption choices of rural residents |
| Material factor | Consumer price index (last year = 100) | Characterizes price fluctuations in rural commodity markets |
| | Food consumption | Refers to the amount of food consumed by rural residents |
| Price factor | Food retail price index (last year = 100) | Reflects the selling price of food in rural areas |

Finally, we took the per capita food consumption carbon emissions as an independent variable, and the economic–preference factor (first principal component), the education–social factor (second principal component), the material factor (third principal component), and the price factor (fourth principal component) as dependent variables to construct a geographically weighted regression model.

3.4.2. Spatial Differentiation of Effective Factors

Through the analysis in Section 3.3, we can see that the rural residents' per capita food consumption carbon emissions in China had spatial autocorrelation characteristics. We ran the geographically weighted regression model (GWR), selected the core type as adaptive, set the bandwidth as AICc, and carried out local spatial autoregressive analysis (Table 4). We concluded that the standardized residual ranges of the local regression models in each province in 2002 and 2020 were [−3.01,2.41] and [−2.29,2.35], respectively, both of which are lower than 2.5. Further global autocorrelation analysis was performed to obtain Moran's $I$ = 0.017, $Z$ = 0.470, and $P$ = 0.639 in 2002, and Moran's $I$ = −0.168, $Z$ = −1.235, and $P$ = 0.217 in 2020. These values show that the standardized residuals were

distributed completely randomly in space, that is, the GWR model better revealed the spatial distribution characteristics of the driving factors of per capita food consumption carbon emissions.

**Table 4.** Regression results of the GWR model.

| Parameters | 2002 Year | 2020 Year |
|---|---|---|
| Neighbors | 29 | 29 |
| Residual squares | 17,872.616 | 41,339.173 |
| Effective number | 10.930 | 11.154 |
| Sigam | 29.841 | 45.640 |
| AICc | 315.677 | 342.530 |
| R square | 0.833 | 0.802 |
| Adjusted R square | 0.750 | 0.700 |

The results in Table 4 indicate that this article considers the GWR model to be reasonable to a certain degree. Therefore, the GWR model was used to analyze the spatial heterogeneity of the influencing factors of rural residents' per capita food consumption carbon emissions. The main contents are as follows.

Economic–Preference Factors Impact Analysis

We combined Formula (5) to run the GWR model to calculate the regression coefficients of each driving factor, and divided it into four levels from low to high according to the natural break point analysis method (Jenks). Then, 2002 and 2020 were selected for visualization for the purpose of revealing the spatial heterogeneity of independent variables on dependent variables (Figures 4 and 5). The results show that there was a positive correlation between the economic–preference factor and per capita food consumption carbon emissions in 2020, and the regression coefficient was higher in the west and lower in the south. Among them, Xinjiang, Tibet, and Qinghai were high-value agglomeration areas. According to the global development trend in food consumption, with improvements in economic levels, the consumption of animal-based food by residents will continue to increase [36], which will promote the increase in their food consumption carbon emissions [21]. During the study period, economic strength and residents' income and expenditure levels in these areas were significantly improved, resulting in the influence of the economic–preference factor on the region being far greater than in other provinces.

The areas with low regression coefficients were mainly concentrated in the south area of the Huaihe River and Qinling Mountains, indicating that the rural residents' food consumption in this area has a relatively small impact on the environment. The environmental Kuznets curve showed that the regional environmental quality will show an "inverted U-shaped" development trend with improvements in economic level. In 2020, the growth rate of per capita GDP and per capita disposable income among rural residents in low-value areas was not only higher than that of other regions, but also higher than that of first-tier cities such as Beijing and Shanghai; however, their per capita food consumption carbon emissions also increased, indicating that it is still at the left end of the environmental Kuznets curve and the inflection point has not yet appeared. In summary, the economic–preference factor has a significant role in promoting the growth of per capita food consumption carbon emissions, and the impact is higher in the west and lower in the south.

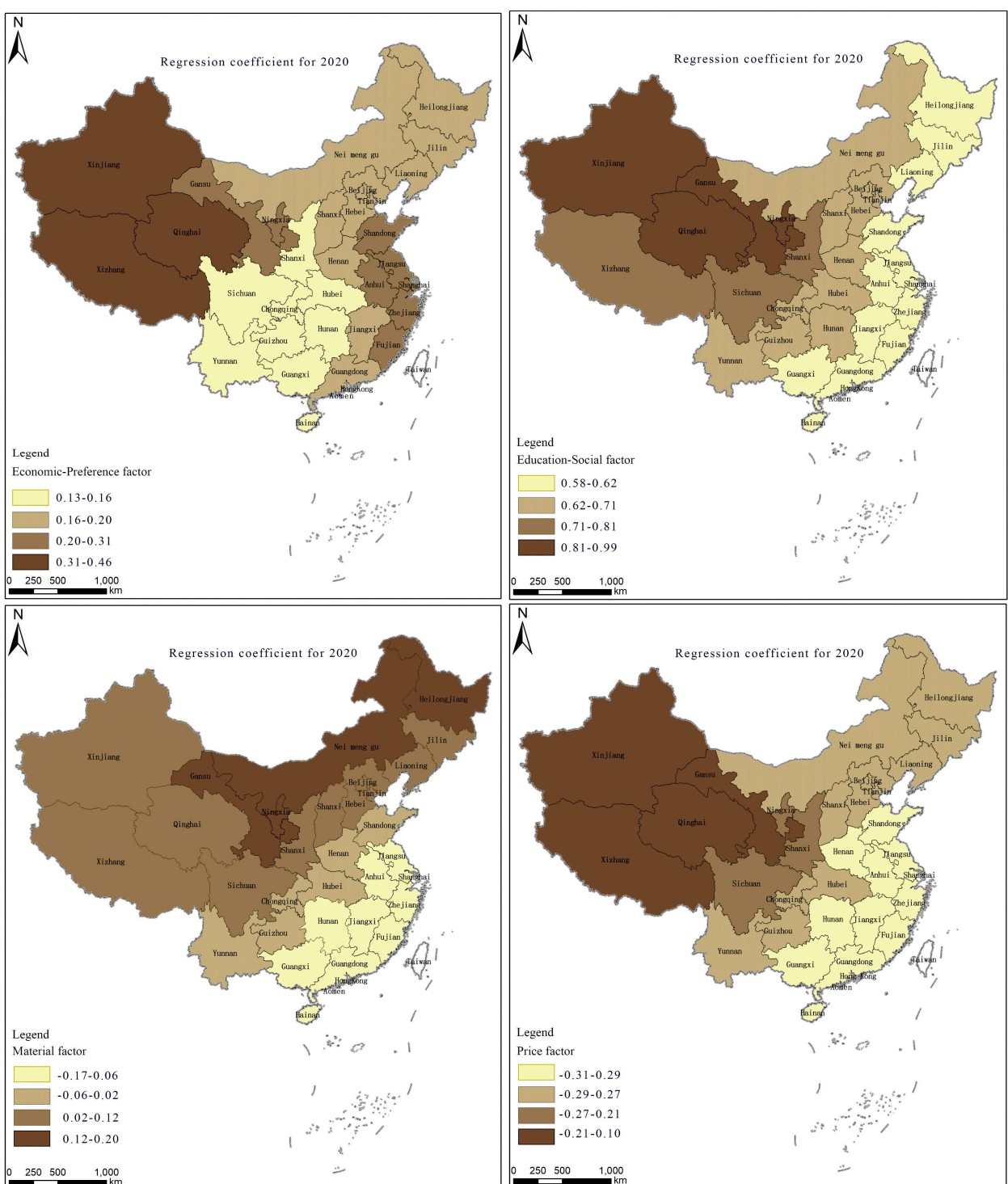

**Figure 4.** Spatial distribution of regression coefficients in 2020.

Education–Social Factor Impact Analysis

The education–social factor showed a positive correlation with per capita food consumption carbon emissions, and the regression coefficients decreased from west to east. High-value areas were concentrated in the western region, where the rural residents' education level continuously improved and the Engel coefficient continuously decreased. Some studies have shown that the higher the level of rural residents' education, the stronger their awareness is of environmental protection [37]. At the same time, the diversification of consumption choices also makes them pay more attention to the pursuit of spiritual life [38],

which means that there should have been a negative correlation between the education–social factor and carbon emissions of per capita food consumption. However, we found a different conclusion. Although the number of years in education gradually increased, it is still at the stage of a nine-year compulsory education. People with higher education rather than compulsory education are the first to come into contact with pollution-free food [39]; therefore, the level of education conferred in junior high school cannot inhibit the growth of per capita food consumption carbon emissions. At the same time, although the Engel coefficient of rural residents in the four provinces continuously decreased, the expenditure on food consumption increased by 413.07% on average, which is higher than the national average growth rate of 349.05%. Therefore, the education–social factor has a role in promoting the growth of per capita food consumption carbon emissions, and its influence decreases from west to east.

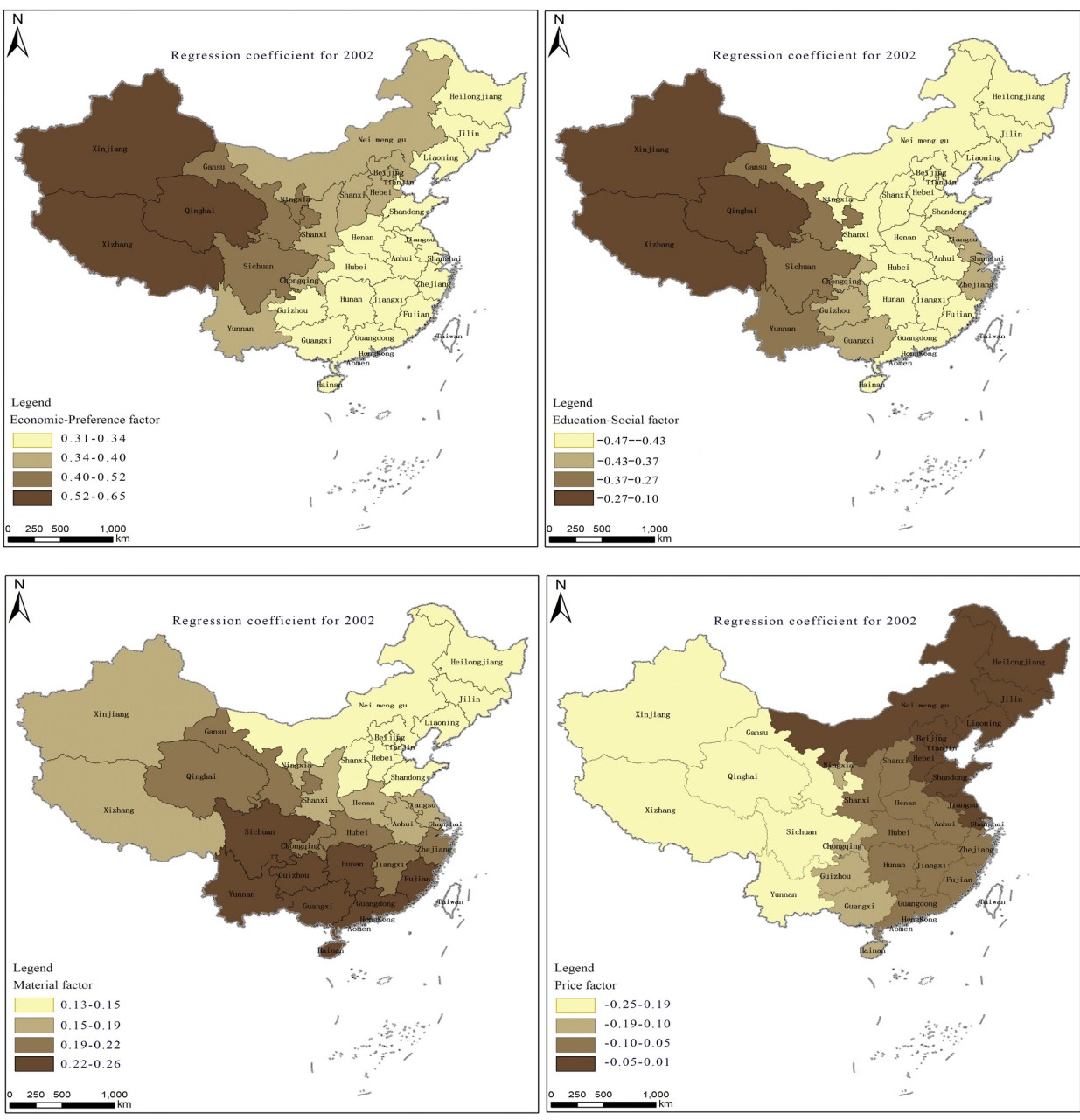

**Figure 5.** Spatial distribution of regression coefficients in 2002.

Material Factor Impact Analysis

The regression coefficient of the material factor had both positive and negative factors, decreasing from north to south and indicating the instability of its influence. The positive-value regions were mainly distributed in northern regions such as Heilongjiang and Inner Mongolia. The consumer price increase in these areas was weak, and has little impact on food purchasing power, whereas high food consumption creates high carbon emissions. The negative-value areas were mainly distributed in the southeastern provinces. After long-term development, the rural commodity market has gradually improved in this area. The increase in consumer prices reduces carbon emissions by curbing rural residents' food purchases. This is consistent with the existing research results of Wang et al. (2010) [38].

Price Factor Impact Analysis

The price factor showed a negative correlation with per capita food consumption carbon emissions, and the regression coefficients decreased from northwest to southeast. When other conditions remain the same, the quantity of goods demanded will decrease as their prices rise according to the demand theorem. That said, food is a necessary product to maintain the survival of residents and demand price elasticity is relatively weak. However, the consumption of beef and mutton is high-end consumption, which is greatly affected by its own price [40]. Compared with the northwest inland provinces, the southeastern provinces, where the negative high values were concentrated, have seen a larger increase in food retail prices in recent years. Although their residents' income levels are relatively high, the excessively high prices still restrict their purchase of food and further promote a reduction in per capita food consumption carbon emissions. This is consistent with previous research [41]. Areas with negative or low values were distributed in the inland northwest, where the rural economic foundation is weak and income level is low. Rural residents living in this area are more sensitive to price changes. With the increase in food retail prices, the amount of food purchased by rural residents has been reduced to some extent, indirectly promoting a reduction in their per capita food consumption carbon emissions. Previous studies have also shown that an increase in food retail prices can help reduce the carbon emissions of rural residents [42].

### 3.4.3. Spatial–Temporal Evolution of Influencing Factors

Through a comparison in Figures 4 and 5, we found that high-value regions of the regression coefficient of the economic–preference factor were mainly distributed in Xinjiang, Tibet, and Qinghai from 2000 to 2020, indicating that the impact of this factor on the per capita food consumption carbon emissions is relatively stable. In order to achieve low-carbon food consumption, the dietary structure should be adjusted to reduce the consumption of animal-based foods such as pork, beef, and lamb and increase the consumption of low-fat and high-protein foods such as milk. It is necessary to reduce red meat consumption for the purpose of climate action [43]. If a sustainable diet is adopted, e.g., eating fewer animal-based foods, carbon emissions can be reduced by up to 75% [9,44,45]. The high-value area of the regression coefficient of the education–social factor increased, mainly in the northwest region, but the regression coefficient changed from negative to positive, indicating that the effect of this factor on the growth of per capita food consumption carbon emissions changed from restraining to promoting. It is necessary to continue to increase investment in education and increase the number of years of education of residents in the future so as to stimulate the effect of education on environmental protection [37,46] and reduce residents' excessive food consumption habits [38,41]. The high-value area of the regression coefficient of the material factor shifted from south to north. In order to reduce the carbon emissions of food consumption, it is necessary to improve the operation system of the rural market economy and rely on market forces to restrain residents' unreasonable food consumption and achieve the goal of low-carbon food consumption. Reducing unreasonable food consumption can reduce greenhouse gas emissions in the UK and the EU by 3% and 10%, respectively [47]. Unreasonable food consumption is widespread in China,

causing per capita carbon emissions to increase by 40 kg per year [18]. This is an alarming figure considering China's huge population. The high-value area of the price factor is shifting from the northeast to the northwest and the quantity is declining. In the future, on the basis of ensuring the stability of food retail prices, the price lever should restrain the growth of per capita food consumption carbon emissions in the region. Through reasonable price-control measures, it can significantly promote a reduction in residents' food-related carbon emissions [42,48].

## 4. Discussion

The main objective of our paper was to explore the spatial and temporal evolution of rural residents' food consumption carbon emissions and the spatial heterogeneity of influencing factors from two perspectives: the calculation of carbon emissions generated from food consumption and the analysis of spatial and temporal evolution characteristics, such as the spatial heterogeneity of influencing factors of food consumption carbon emissions. Our results confirm that the carbon emissions generated from rural residents' food consumption is increasing, showing the distribution characteristics of a north–south confrontation, with a central area collapse, and there is significant spatial heterogeneity in the effect of each influencing factor. The research conclusions of this article are consistent with previous studies on the carbon emissions generated from Chinese residents' food consumption. For instance, He et al. (2018) found that greenhouse gas emissions related to food consumption showed an overall increasing trend [49]. Cao et al. (2020) pointed out that the areas with higher carbon emissions are Inner Mongolia, Chongqing, Sichuan, Xinjiang, and Tibet, all of which are located in the northern and southern regions of China. The areas with lower carbon emissions are Shanxi, Henan, Hebei, and Gansu, mainly located in central China [21]. As expected, the empirical results of this paper show that the carbon emissions of rural residents' food consumption increase by 1.68% annually. The provinces with high carbon emissions are Inner Mongolia, Chongqing, Sichuan, Qinghai, Guangdong, Fujian, and Shanghai, whereas Shanxi, Henan, Hebei, Gansu, Shaanxi, and Shandong are provinces with low carbon emissions. Our research results are almost in line with the results of the above-mentioned scholars. However, the conclusions of this article on influencing factors are somewhat different from previous studies. For example, previous studies found that there was a significant positive correlation between level of education and the country's green GDP [50]; people with a good education are more likely to choose environmentally friendly consumption methods [51]. However, we found that the increase in the level of rural residents' education was accompanied by an increase in carbon emissions from food consumption. The reason for the inconsistency of the results may be that residents' low level of education cannot account for the positive role of education in promoting environmental protection [37]. During the study period, the average education level of rural residents stayed at the nine-year compulsory education stage; however, people with higher education rather than compulsory education are the first to come into contact with pollution-free food [39]. Moreover, improvements in education level are accompanied by an increase in the consumption of clothing, transportation and communication, food, culture, education, and entertainment, which in turn increase carbon emissions [52].

The limitation of this article is that it only considers the main types of rural residents' food consumption, and does not include the consumption of cakes, seasonings, or beverages, etc., due to a lack of statistical data; however, this does not affect the generalization of the conclusions. The contributions of this article provide a valuable theoretical basis for policymakers to formulate differentiated low-carbon food consumption policies, thereby achieving sustainable food consumption. Future research will attempt to conduct field surveys in representative areas to obtain microdata on rural residents' food consumption, and to more accurately study the changing trends in carbon emissions from food consumption so as to provide a more accurate reference for the formulation of low-carbon food consumption policies.

## 5. Conclusions

This paper analyzes rural residents' per capita food consumption carbon emissions in 31 provinces in China from 2002 to 2020, and explores the spatial heterogeneity of its driving factors with the help of the ESDA-GWR model. The main conclusions are as follows. First, per capita food consumption carbon emissions of rural residents increased by 1.68% annually to 336.73 kg $CO_2$-eq in 2020, of which animal-based food was the main food type that promoted the growth of per capita carbon emissions. Although the high-value areas of per capita carbon emissions are continuously decreasing, the low-value areas are slowly increasing, which promotes the spatial distribution of rural residents' per capita food consumption carbon emissions to show a north–south confrontation, and a collapse in the middle. Second, the rural residents' per capita food consumption carbon emissions show obvious spatial autocorrelation characteristics. The hot and cold spots were mainly distributed in the southern and northern regions of China, and showed a clear agglomeration effect from 2002 to 2020. Among them, the hot spot areas showed a significant spatial evolution trend, converging to the southeast, and the cold spot areas were relatively fixed. Third, there is significant spatial heterogeneity in the influencing factors of carbon emissions from rural residents' food consumption. The price factor plays the most prominent role in curbing carbon emissions from food consumption, and its role is weakened from northwest to southeast. The education–social factor is the main power driving the growth of food consumption carbon emissions, and its role has been continuously strengthened from east to west.

Based on the above research conclusions, the policy implications are as follows. First, the dietary structure of rural residents needs to be further optimized [20]. In areas with high carbon emissions from food consumption, such as Inner Mongolia, Qinghai, and Sichuan, propaganda power should be increased to enhance residents' attention to scientific diet, reduce residents' intake of meat food, and increase the consumption of dairy products. China has a serious lack of dairy product intake [53]; compared with meat consumption, the impact of increased consumption of dairy products on the environment is far lower than that of meat food consumption [30], and it can also meet the human body's demand for nutrients. Second, we should further improve the education level of rural residents, especially in Xinjiang, Tibet, Qinghai, etc., for the reason that a low level of education might inhibit the promotion of environmental protection [37]; it is also not conducive to residents' pursuit of a spiritual life [38,41]. In the future, investment in education in these areas should be increased, not only to improve "hardware" such as educational infrastructure, but also to increase investment in "software" such as teachers. More outstanding teachers should be encouraged to participate in rural education.

**Author Contributions:** Conceptualization, H.C. and S.Q.; methodology, software, data curation, S.Q. and H.W.; writing—original draft preparation, S.Q.; writing—review and editing, H.W.; supervision, H.C. All authors have read and agreed to the published version of the manuscript.

**Funding:** This research was funded by Heilongjiang Province Ecological Civilization Construction and the Green Development Think Tank Project (grant number 202010).

**Institutional Review Board Statement:** Not applicable.

**Informed Consent Statement:** Not applicable.

**Data Availability Statement:** All data in the paper come from the statistical yearbook compiled by the National Bureau of Statistics of China (http://www.stats.gov.cn/, (accessed on 30 August 2021)); please refer to the second paragraph in Section 2.4 for details. In addition, interested readers can obtain all the data from the corresponding author if required.

**Acknowledgments:** The authors are particularly grateful to Northeast Forestry University, Harbin, China, for their technical support and extend thanks for the project support given by the fund granting unit.

**Conflicts of Interest:** The authors declare no conflict of interest.

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
