# Peer review of "Spatial–Temporal Heterogeneity and Driving Factors of Rural Residents’ Food Consumption Carbon Emissions in China—Based on an ESDA-GWR Model"

_sustainability, doi:10.3390/su132212419_

Round 1
Reviewer 1 Report
This study used China's statistical yearbook data to calculate the per capita food consumption carbon emission. This paper also examines spatial heterogeneity and driving factors of food consumption carbon emission. This is exciting research. However, I do have a few comments about the quality of the presentation of this study. For example, the resolution of equations/ figures can be improved to be easier to follow.
The introduction is too short, and it lacks the objective of this study, contribution, and how this is different from closely related other studies.
The discussion section could be a separate topic that highlights the current study's importance and compares it with the previous finding along with policy implications.
Please make to make all tables and figures stand alone and are in the same format. Also, please maintain consistency throughout the figure. For example, legends of figure 1 should be Egg not egg.
Please remove suggestions. It could be a separate paraph in discussion and a couple of sentences in conclusion.
Author Response
Dear Sir/Madam,
Please see the attachment

Reviewer 2 Report
Dear authors
the paper is interesting and it is adaptable to this journal. However, I see some issues that can be resolved. In particular,
- abstract is not attractived Lines 17-21 there are no direct results
- Line 58 there is an additional point, but all structure can be improved added some relavant concepts: social dimension of sustainability https://doi.org/10.1016/j.jclepro.2021.128960 the role of technology https://doi.org/10.3390/su13179942 sustainable hand https://doi.org/10.3390/en14185661
- Section 1 can define clearly its novelty than literature Lines 62-63 are not sufficient
- Title 2.1 are missing some words? what is the methodology used than literature?
- what are the input data? I don't see clearly this aspect within section 2
- Until Line 299 is ok, after what is 1? Again 328 ,349 , ...please check all document
- what is the role of critical variables? what is the impact of sustainability on these aspects? what is your contribution than previous studies?
- Line 434 is 4.2
- what are the main issues of your work?
Author Response
Dear Sir/Madam,
Please see the attachment.

Reviewer 3 Report
This manuscript by Qin et al focused on the carbon emissions of rural residents' food consumption in China. Compared to previous studies that targeted urban areas, this study helped to fill a previously missing piece of the high-carbon emission from food consumption in China. There is no big flaw in grammar or spelling and the structure of the paper is well-organized. The only concern regarding this paper is that the discussion of price and spatial factors were not compared against any previous work and more proof and details would be helpful to explain the reasons behind determining these factors, making the regression model more solid.
Author Response

(The authors gave the same response as above.)

Round 2
Reviewer 2 Report
congratulations.